# PFKFB3-Mediated Glycolysis Boosts Fibroblast Activation and Subsequent Kidney Fibrosis

**DOI:** 10.3390/cells12162081

**Published:** 2023-08-17

**Authors:** Qiuhua Yang, Emily Huo, Yongfeng Cai, Zhidan Zhang, Charles Dong, John M. Asara, Qingqing Wei

**Affiliations:** 1Department of Cellular Biology and Anatomy, Medical College of Georgia, Augusta University, Augusta, GA 30912, USA; 2Augusta Preparatory Day School, 285 Flowing Wells Rd, Martinez, GA 30907, USA; 3Dental College of Georgia, Augusta University, Augusta, GA 30912, USA; 4Division of Signal Transduction, Beth Israel Deaconess Medical Center and Department of Medicine, Harvard Medical School, Boston, MA 02215, USA

**Keywords:** PFKFB3, kidney, fibrosis, glycolysis, myofibroblast, lactate

## Abstract

Renal fibrosis, a hallmark of chronic kidney diseases, is driven by the activation of renal fibroblasts. Recent studies have highlighted the role of glycolysis in this process. Nevertheless, one critical glycolytic activator, 6-phosphofructo-2-kinase/fructose-2,6-bisphosphatase 3 (PFKFB3), remains unexplored in renal fibrosis. Upon reanalyzing the single-cell sequencing data from Dr. Humphreys’ lab, we noticed an upregulation of glycolysis, gluconeogenesis, and the TGFβ signaling pathway in myofibroblasts from fibrotic kidneys after unilateral ureter obstruction (UUO) or kidney ischemia/reperfusion. Furthermore, our experiments showed significant induction of PFKFB3 in mouse kidneys following UUO or kidney ischemia/reperfusion. To delve deeper into the role of PFKFB3, we generated mice with *Pfkfb3* deficiency, specifically in myofibroblasts (*Pfkfb3*^f/f^/*Postn*^MCM^). Following UUO or kidney ischemia/reperfusion, a substantial decrease in fibrosis in the injured kidneys of *Pfkfb3*^f/f^/*Postn*^MCM^ mice was identified compared to their wild-type littermates. Additionally, in cultured renal fibroblast NRK-49F cells, PFKFB3 was elevated upon exposure to TGFβ1, accompanied by an increase in α-SMA and fibronectin. Notably, this upregulation was significantly diminished with PFKFB3 knockdown, correlated with glycolysis suppression. Mechanistically, the glycolytic metabolite lactate promoted the fibrotic activation of NRK-49F cells. In conclusion, our study demonstrates the critical role of PFKFB3 in driving fibroblast activation and subsequent renal fibrosis.

## 1. Introduction

Chronic kidney disease (CKD) is a major renal disease that affects a large population worldwide and imposes significant socio-economic burdens. Regardless of the underlying causes, renal fibrosis emerges as a prominent pathology that ultimately leads to the deterioration of renal function [1]. Renal fibrosis can be triggered by renal tubular epithelial injury, regulated by the infiltration of inflammatory cells such as macrophages, and driven by myofibroblast proliferation [2,3,4,5]. Myofibroblasts, a specific subtype of activated fibroblasts characterized by the expression of alpha-smooth muscle actin (α-SMA), play a vital role in the progression of renal fibrosis. Various cell types have been identified as potential sources of myofibroblasts during the development of renal fibrosis, including resident fibroblasts, hematopoietic fibrocytes, and pericytes [6,7,8]. In renal fibrosis, myofibroblasts are predominantly responsible for the excessive production and secretion of collagen, fibronectin, and other extracellular matrix components [6,9].

Recent studies have highlighted the importance of metabolic reprogramming, particularly the induction of glycolysis, in the progression of renal fibrosis [10]. Increased glycolytic metabolites have been found in obstructive kidneys [11], maladaptive kidneys after ischemic injury [12], and kidneys with diabetic nephropathy (DN) [13]. In mouse unilateral ureter obstruction (UUO) kidneys, Ding et al. showed an increased expression of PKM2, a key enzyme of glycolysis [14]. Both PKM2 knockdown and glycolysis inhibition by shikonin or 2-deoxyglucose suppressed renal fibrosis in obstructive kidneys [14]. In vitro, in cultured renal fibroblasts, knockdown of PKM2 inhibited the proliferation of myofibroblasts and the production of fibronectin [14]. In our previous study [15], we also confirmed the role of glycolysis in renal fibrosis with glycolysis inhibitor treatment, such as shikonin or dichloroacetate (DCA), in UUO models. However, glycolysis inhibition showed a divergent effect on renal epithelial cells and renal fibroblasts. While shikonin and DCA can inhibit fibronectin and α-SMA production in renal fibroblasts, we mainly identified the suppression of renal tubular cell death by glycolysis inhibition in UUO. Considering the variety of renal cells contributing to renal fibrosis, it is important to further explore the specific role of glycolysis in fibroblasts.

6-Phosphofructo-2-kinase/fructose-2,6-bisphosphatase 3 (PFKFB3) is an activator of glycolysis. PFKFB3 catalyzes the synthesis of fructose-2, 6-bisphosphate (F-2,6-P2), while F-2,6-P2 is the most potent allosteric activator of 6-phosphofructo-1-kinase (PFK-1), one of three rate-limiting enzymes for glycolysis [16]. PFKFB3 is upregulated in proliferative cells, vascular cells, and immune cells under the conditions of inflammation and stresses [17]. *Pfkfb3* deficiency in vascular cells and myeloid cells protected mice from developing pulmonary hypertension [18], ocular angiogenesis [19,20], and atherosclerosis [21]. The role of PFKFB3 in fibrosis was primarily explored in an experimental model of pulmonary fibrosis [22], in which PFKFB3 knockdown and an inhibitor suppressed the fibrotic activity of lung myofibroblasts and lung fibrosis in a mouse model. In kidneys, we reported PFKFB3 induction in cisplatin-induced acute kidney injury and found that its suppression ameliorates cisplatin nephrotoxicity [23]. However, it remains unclear whether PFKFB3 in renal fibroblasts is crucial for the fibrotic activity of myofibroblasts and subsequent renal fibrosis.

In this study, to determine the specific role of glycolysis in myofibroblasts and renal fibrosis, we generated mice with *Pfkfb3* deficiency, specifically in myofibroblasts, and challenged them with ureter obstruction or kidney ischemia to induce fibrosis. Combined with in vitro assays to examine PFKFB3 knockdown and inhibition in renal fibroblasts, we elucidated the pro-fibrotic role of PFKFB3 in myofibroblasts and its role through downstream glycolysis regulation in renal fibrosis.

## 2. Materials and Methods

### 2.1. Single-Cell Data Sequencing Analysis

Single-cell RNA sequencing data were downloaded and all the cells were combined for UUO and kidney ischemia/reperfusion models, respectively [24]. Genes for different pathways from the WikiPathway database (https://www.wikipathways.org/) (accessed on 13 June 2023) were grouped for glycolysis, gluconeogenesis, and the TGFβ signaling pathway. Each cell from fibroblast or myofibroblast clusters was scored using the AUCell package (Laboratory of Computational Biology. VIB-KU Leuven Center for Brain & Disease Research. Leuven, Belgium) with a threshold of 0.01 [25]. Data were analyzed with RStudio 2023.06.0 + 421 (RStudio: Integrated Development for R. RStudio Inc., Boston, MA, USA).

### 2.2. Animals

All animal care and experimental procedures were conducted following the protocol approved by the Institutional Animal Care and Use Committee at Augusta University. Briefly, the mice were housed in an environment with controlled temperature and humidity and a 12-h/12-h light/dark cycle. All animals had ad libitum access to food (Teklad global 18% protein rodent diet; 2918-060622M, Envigo, Madison, WI, USA) and water. Floxed *Pfkfb3* (*Pfkfb3*^flox/flox^) mice (generated by Xenogen Biosciences Corporation, Cranbury, NJ, USA) [26] were crossbred with *Postn*-Cre transgenic mice (The Jackson Laboratory, stock no. 029645, Bar Harbor, ME, USA) to generate myofibroblast-specific *pfkfb3*-knockout (*Pfkfb3*^f/f^/*Postn*^MCM^) mice, with *Pfkfb3*^WT/WT^
*Postn*-Cre-positive (*Postn*^MCM^) mice used as wild-type controls. The knockout of *Pfkfb3* in myofibroblasts was induced with a tamoxifen injection (75 mg/kg/day for 5 days, i.p.), followed by tamoxifen-containing mouse chow diet treatment (TD.130860, Envigo, Indianapolis, IN, USA). The mouse genotype was confirmed using PCR amplification of tail-clip samples (Appendix A).

### 2.3. Unilateral Ureteral Obstruction (UUO) Mouse Model

UUO surgery was conducted, as described previously [15]. Briefly, adult male mice weighing between 20 and 25 grams were anesthetized with ketamine (100 mg/kg) and xylazine (10 mg/kg) (i.p. injection). A laparotomy incision was made to expose the left ureter. The ureter was obstructed with double ligation with silk suture and a cut in the ureter made in between. After 2 weeks of UUO, the mice were euthanized with ketamine (100 mg/kg) and xylazine (10 mg/kg), and the kidney samples were fixed with 4% paraformaldehyde for histological examination or snap-frozen in liquid N_2_ for protein and gene expression analysis. Contralateral kidneys without ligation were used as controls.

### 2.4. Unilateral Kidney Ischemia/Reperfusion Mouse Model

Kidney ischemia/reperfusion surgery was performed, as described before [27]. Adult male mice were given ketamine (100 mg/kg) and xylazine (10 mg/kg) (i.p. injection) for anesthesia. Surgery was performed on a homeothermic blanket control unit with a rectal probe (507220F, Harvard Apparatus, Holliston, MA, USA) to monitor and maintain the body temperature at ~36.5 °C. A flank incision was made to expose the left renal pedicle, which was clamped for 30 min before releasing for reperfusion. The kidney color change during renal pedicle clamping was monitored to ensure the success of ischemia and reperfusion. The mice were sacrificed at 2 weeks of reperfusion to collect kidneys for histological examination and protein and gene expression analysis. Contralateral kidneys without ischemia/reperfusion were used as controls.

### 2.5. Cells

Normal rat kidney interstitial fibroblast (NRK-49F) cells were purchased from the American Type Culture Collection (Manassas, VA, USA) and cultured in DMEM with 10% FBS (F4135, Sigma-Aldrich, St. Louis, MO, USA) in a humidified incubator with 5% CO_2_ at 37 °C. For testing TGFβ1-induced fibroblast activation, cells were incubated in serum-free DMEM with a vehicle or recombinant human TGFβ1 (100-B-010-CF; R&D, Minneapolis, MN, USA) and lactate (Sigma-Aldrich, St. Louis, MO, USA), as indicated in figures. AZ-26 (HY-101971, MedChemExpress, Monmouth Junction, NJ, USA), a selective PFKFB3 inhibitor, was added to serum-free medium for 30 min before TGFβ1 treatment. To knock down PFKFB3, cells were transfected with *Pfkfb3* siRNA (L-095107-02-0005, Dharmacon, Lafayette, CO, USA) using lipofectamine NRAiMAX (13778150, Invitrogen, Grand Island, NY, USA) according to the manufacturer’s instructions and compared to cells transfected with the negative control siRNA (D-001810-10-05, Dharmacon).

### 2.6. Reverse Transcription and Quantitative Real-Time PCR (RT-qPCR)

Trizol reagent (15596018, Invitrogen, Grand Island, NY, USA) was used for total RNA extraction following the manufacturer’s protocol [28]. cDNAs were synthesized with the iScriptTM cDNA synthesis kit (1708891, Bio-Rad Hercules, CA, USA). RT-qPCR was performed on the StepOne Plus System (Applied Biosystems, Grand Island, NY, USA) with the Power SYBR Green Master Mix (1725122, Bio-Rad Hercules, CA, USA). The relative gene expression was quantified using the efficiency-corrected 2^−△△CT^ method, with 18S ribosomal RNA serving as the internal control. The data were presented as fold change relative to the control groups. The primers used for RT-qPCR are listed in Appendix A.

### 2.7. Western Blot Analysis

Total kidney protein extracts or cell lysates were prepared in RIPA buffer (PI89900, Fisher Scientific, Pittsburgh, PA, USA) supplemented with protease inhibitor cocktails (05892970001, Roche, SC, USA). The kidney tissues were ground and then lysed to prepare tissue lysates. The cultured cells were directly lysed with RIPA buffer in a dish. The extracts were centrifuged at 12,000 rpm for 10 min, and the resulting supernatant was collected. The protein concentration was determined using the Rapid Gold BCA Protein Assay Kit (PIA53225, Fisher Scientific, Pittsburgh, PA, USA). Subsequently, the samples were subjected to SDS-PAGE, transferred onto nitrocellulose membranes, and incubated with primary antibodies after blocking with 5% skim milk. The following primary antibodies were used: PFKFB3 (ab181861, abcam, Waltham, MA, USA), ACTA2 (sc-56499, Santa Cruz Biotechnology, Dallas, TX, USA), Vimentin (CST5741, Cell Signaling Technology, Beverly, MA, USA), Collagen I (NB600-408, NOVUS Biologicals, Centennial, CO, USA), Fibronectin (ab2413, abcam, Waltham, MA, USA), Cyclophilin B (CST43603, Cell Signaling Technology, Beverly, MA, USA), PCNA (sc56, Santa Cruz Biotechnology, Dallas, TX, USA), anti-β-actin (sc47776, Santa Cruz Biotechnology, Dallas, TX, USA), anti-p-Smad2 (S465/467) (3208S, Cell Signaling Technology, Beverly, MA, USA), anti-Smad2 (3122S, Cell Signaling Technology, Beverly, MA, USA), and a-tubulin (#3873, Cell Signaling Technology, Beverly, MA, USA). The enhanced chemiluminescence signal was detected and quantified, as described before [29]. The antibodies used are listed in Appendix A.

### 2.8. Immunofluorescence

Cells were cultured in 8-chambered cell culture slides (08-774-26, Fisher Scientific, Pittsburgh, PA, USA), or frozen or paraffin-embedded sections of 7 μm thickness were used for staining. Cells or frozen sections were fixed with 4% paraformaldehyde for 15 min and permeabilized using 0.5% Triton X-100 for 20 min at room temperature. The paraffin-embedded sections were subjected to deparaffinization and rehydration with xylene/ethanol serial incubation. For antigen retrieval, slides were heated in sodium citrate buffer (10 mM, pH 6.0) by microwaving at 98 °C for 10 min. Next, the cells or sections were blocked with 10% normal goat serum (50062Z, Thermo Scientific, Waltham, MA, USA) for 1 h at room temperature, followed by incubation with a primary antibody overnight at 4 °C. Secondary antibodies conjugated with Alexa Fluor 488, 594, or 647 were used to detect specific signals. Nuclei were counterstained with DAPI for 10 min at room temperature. Images were taken with an inverted confocal microscope (Zeiss 780, Carl Zeiss, Oberkochen, Germany) and quantified with a method described previously [19]. The antibodies used are listed in Appendix A.

### 2.9. Immunohistochemistry

Paraffin-embedded kidney sections were deparaffinized, rehydrated, and treated with methanol containing 30% H_2_O_2_ for 30 min at room temperature. After antigen retrieval with 10 mM sodium citrate buffer (pH 6.0), the sections were blocked with avidin blocking solution for 1 h at room temperature, followed by incubation with a primary antibody overnight at 4 °C. Primary antibodies were mixed with biotin blocking solution (SP-2001, Vector Laboratories, Newark, CA, USA) as per the manufacturer’s instructions. Subsequently, the tissue sections were treated with a biotinylated secondary antibody for 1 h at room temperature, followed by incubation with ABC solution (PK-6100, Vector Laboratories, Newark, CA, USA) for 30 min at room temperature. The signal was visualized using the peroxidase substrate 3,3’-diaminobenzidine (3468, Dako, Santa Clara, CA, USA), and the nuclei were counterstained with hematoxylin Ⅰ (GHS116, Sigma, St. Louis, MO, USA ). Finally, the slides were dehydrated and mounted using a xylene-based mounting medium (8312-4, Richard-Allan Scientific, San Diego, CA, USA). Images were taken with a Keyence Microscope (BZ-X800 Keyence Corporation of America, Itasca, IL, USA) and quantified with a method described previously [30]. The antibodies used are listed in Appendix A.

### 2.10. Histological Staining

The paraffin-embedded sections were stained with hematoxylin (22050111, Fisher Scientific, Pittsburgh, PA, USA) and eosin (220501110, Fisher Scientific, Pittsburgh, PA, USA) (H&E) following the manufacturer’s protocol. Collagen deposition was detected using Masson’s trichrome staining kit (25088-1, Polysciences, PA, USA) and the Sirius Red staining kit (ab150681, Abcam, CA, USA) according to the manufacturer’s instructions. Four random images in the cortex/outer medulla region per section were captured with a Keyence Microscope BZ-X800 (KEYENCE, Itasca, IL, USA), and the percentage of the collagen deposition area was quantified with Image J software (National Institutes of Health, Bethesda, MD, USA, http://imagej.nih.gov/ij).

### 2.11. Metabolomics Assay

Biological triplicate 10 cm^2^ dishes were used to cultivate NRK-49F cells in full culture medium. Metabolites were extracted using 1 mL of ice-cold 80% methanol on dry ice. Subsequently, the samples were centrifuged at 14,000 rpm for 5 min. To ensure thorough extraction, the cell pellets were subjected to an additional extraction with 0.5 mL of 80% methanol. For accurate protein quantitation, the cell pellets were dissolved in an 8 M urea solution. The supernatant obtained from the metabolite extraction was desiccated into a pellet using SpeedVac from Eppendorf (Hamburg, Germany), using a heat-free technique. Before analysis, the dried pellets were re-suspended in 20 μL of HPLC-grade water in preparation for mass spectrometry, as described before [31]. A volume of 5–7 μL of the resulting resuspension was injected and subjected to analysis using a cutting-edge hybrid 6500 QTRAP triple quadrupole mass spectrometer from AB/SCIEX (Framingham, MA, USA), which was coupled to a Prominence UFLC HPLC system from Shimadzu (Kyoto, Japan). The analysis was carried out through selected reaction monitoring (SRM), targeting a comprehensive set of 298 endogenous water-soluble metabolites, enabling a thorough examination of the steady-state characteristics of the samples. The data have been deposited to MetaboLights (MTBLS8281) [32].

### 2.12. Statistical Analysis

Statistical analysis was performed with GraphPad Prism software (version 9.0, RRID: SCR_000306, GraphPad Software, Boston, MA, USA). The difference between two groups was evaluated using the unpaired Student’s *t*-test or the unpaired two-tailed Student’s *t*-test with Welch’s correction (with unequal variances). Multiple comparisons were performed through one-way ANOVA, followed by the Bonferroni post hoc test. All results are presented as the mean ± SEM. A *p*-value of <0.05 was considered statistically significant. Each biological experiment was repeated a minimum of three times, using independent cell cultures or individual animals as biological replications.

## 3. Results

### 3.1. Induction of PFKFB3 Expression in Fibrotic Kidneys

PFKFB3 is an enzyme that promotes glycolysis metabolism. Thus, we assessed its expression in mouse fibrotic kidneys. In normal kidneys, the expression level of PFKFB3 was found to be low, with minimal protein expression detected (Figure 1A,B). However, in kidneys subjected to 2 weeks of UUO, both the mRNA and protein levels of PFKFB3 were significantly increased (Figure 1A,B). Similarly, in kidneys subjected to 30 min of ischemia followed by 2 weeks of reperfusion, PFKFB3 expression induction was also observed (Figure 1C,D).

### 3.2. Increased Glycolysis and TGFβ Pathways in Renal Myofibroblasts

To gain insight into the regulation of glycolysis in the process of fibroblast activation, we conducted a reanalysis of single-cell sequencing data obtained from Dr. Humphreys’ lab [24]. All the data from kidneys subjected to UUO (2, 4, 6, 10, and 14 days) or ischemia with reperfusion (6 h, 2, 7, 14 and 28 days) were analyzed together, allowing for a comparison of gene expression between fibroblasts and myofibroblasts. We observed a significant enhancement of gene expression related to glycolysis and gluconeogenesis pathways in these myofibroblasts compared to fibroblasts (Figure 2A,B). Consistent with the fibrotic nature of these models, genes involved in the TGFβ pathway, which is a major signaling pathway known to promote the activation of fibroblasts into myofibroblasts, were significantly increased in myofibroblasts compared to fibroblasts in both injury models (Figure 2A,B).

### 3.3. Myofibroblast-Specific Pfkfb3 Knockout Reduces Kidney Atrophy and Renal Fibrosis following UUO

To further examine the role of PFKFB3 in renal fibroblast activation, we established mouse models with *Pfkfb3*-inducible knockout in myofibroblasts (*Pfkfb3*^f/f^/*Postn*^MCM^). Both *Pfkfb3*^f/f^/*Postn*^MCM^ and their wild-type littermates *Postn*^MCM^ mice were subjected to UUO for 2 weeks with tamoxifen to induce the knockout of *Pfkfb3* in myofibroblasts (Figure 3A). The animal weight loss after UUO surgery was similar, except that *Pfkfb3*^f/f^/*Postn*^MCM^ mice had less weight loss only on the third day of UUO (Figure 3B). Compared to the control kidneys, the obstructive kidneys experienced obvious weight loss after 2 weeks of UUO, which was significantly blocked by myofibroblast *Pfkfb3* knockout (Figure 3C,D).

To examine whether this kidney atrophy was accompanied with renal fibrosis differences, we tested the protein or mRNA expression of multiple renal fibrosis markers. UUO injury significantly induced the mRNA expression of *Acta2, Collagen I, Collagen III, Mmp2*, and *Mmp9*, while this induction was remarkably suppressed in *Pfkfb3*^f/f^/*Postn*^MCM^ kidneys compared to *Postn*^MCM^ kidneys (Figure 3E). At the protein level, the induction of the myofibroblast marker ACTA2 and extracellular matrix components collagen I and fibronectin were all repressed by myofibroblast *Pfkfb3* knockout (Figure 3F). Of note, Vimentin induction in UUO kidneys, which indicates not only fibroblast activation but also the degeneration of renal tubular epithelial cells, was inhibited after myofibroblast *Pfkfb3* knockout (Figure 3F).

To further confirm the amelioration of renal injury and fibrosis by myofibroblast *Pfkfb3* knockout, we examined the kidney histological morphology with H&E staining. After 2 weeks of UUO, the wild-type kidneys showed massive renal tubular dilation and thinning, which was clearly suppressed in the knockout kidneys (Figure 4A). Furthermore, the interstitial fibrosis in UUO kidneys was attenuated in *Pfkfb3*^f/f^/*Postn*^MCM^ kidneys compared to Postn^MCM^ kidneys, as shown by the significant differences in collagen deposition upon Masson’s trichrome staining, Sirius Red staining, and the immunostaining of Collagen I and collagen IV (Figure 4B,C). The ACTA2 staining proved the blocked activation of fibroblasts by *Pfkfb3* knockout (Figure 4C).

### 3.4. Myofibroblast-Specific Pfkfb3 Knockout Suppresses Kidney Atrophy and Fibrosis after Kidney Ischemia/Reperfusion

To verify the pro-fibrotic role of myofibroblast PFKFB3, we further challenged *Pfkfb3*^f/f^/*Postn*^MCM^ and *Postn*^MCM^ mice with unilateral 30 min of kidney ischemia and 2 weeks of reperfusion (UI30/2wk) (Figure 5A). No obvious weight loss difference was detected in these mice after surgery, suggesting that they experienced similar initial acute injury. Similarly, as in the UUO model, kidney atrophy was partially rescued by myofibroblast *Pfkfb3* knockout (Figure 5C,D). The renal fibrosis amelioration was confirmed by the reduction in mRNA levels of *Acta2, Collagen I, Collagen III, Mmp2*, and *Mmp9* (Figure 5E) and the protein levels of ACTA2, fibronectin, and Collagen I (Figure 5F) in UI30/2wk *Pfkfb3*^f/f^/*Postn*^MCM^ kidneys compared to *Postn*^MCM^ kidneys. Moreover, the interstitial collagen deposition (Masson’s trichrome staining, Sirius Red staining, and immunostaining of collagen I and collagen IV) also indicated significant suppression of renal fibrosis by myofibroblast *Pfkfb3* knockout (Figure 6B,C). The Vimentin induction change reflected both fibroblast activation and tubular degeneration decrease with *Pfkfb3* knockout (Figure 5F). In addition, in UI30/2wk kidneys, it was interesting to note that myofibroblast *Pfkfb3* knockout showed a protective effect on tubular dilation upon H&E staining (Figure 6A).

### 3.5. Increased PFKFB3 Expression in TGFβ1-Activated Renal Fibroblasts

The TGFβ signaling pathway is a primary pathway involved in inducing fibroblast activation [33]. In order to examine the role of PFKFB3 regulation in fibroblast activation, we treated cultured renal fibroblasts (NRK-49F cells) with TGFβ1. TGFβ1 treatment significantly increased both ACTA2 mRNA and protein levels in a concentration-dependent manner (Figure 7A,B) and a time-dependent manner (Figure 7C,D), indicating fibroblast activation (Figure 7). Furthermore, TGFβ1 treatment also led to a significant increase in PFKFB3 mRNA and protein levels (Figure 7A–D). The correlation between TGFβ1-induced PFKFB3 and ACTA2 induction was confirmed with co-immunostaining of the proteins (Figure 7E). Importantly, PFKFB3 could be induced with a lower concentration of TGFβ1 or a shorter treatment time compared to ACTA2, suggesting that PFKFB3 acts as an upstream regulator of fibroblast activation.

### 3.6. PFKFB3 Induction Promotes TGFβ1-Induced Renal Fibroblast Activation

To investigate the regulatory role of PFKFB3 in fibroblast activation, we used siRNA to knock down *PFKFB3* in NRK-49F cells (Figure 8A). In cells transfected with negative control siRNA, fibroblast activation and the expression of fibrosis markers, including ACTA2, fibronectin (FN), and a proliferation marker, proliferating cell nuclear antigen (PCNA), were significantly induced upon TGFβ1 treatment. However, all these inductions were effectively suppressed upon *PFKFB3* knockdown (Figure 8B,C). Moreover, we observed that the knockdown of *PFKFB3* has a reciprocal impact on the activation of the TGFβ1 signaling pathway, as evidenced by the notable decrease in Smad2 phosphorylation (Figure 8D), though the mechanism is unknown. To further confirm the functional significance of PFKFB3, we used its specific inhibitor, AZ-26. Treatment with AZ-26 resulted in the efficient suppression of ACTA2, FN, and PCNA induction by TGFβ1, and this suppression exhibited a concentration-dependent response (Figure 8E–G).

### 3.7. PFKFB3-Medicated Glycolysis Induction Is Responsible for Renal Fibroblast Activation

Since PFKFB3 is known to be an activator of the glycolysis pathway, we conducted further investigations to explore the role of glycolysis-related metabolic changes by PFKFB3 in renal fibroblast activation. Metabolomics analysis revealed relatively low levels of multiple glycolysis metabolites in normal NRK-49F cells, which significantly increased by TGFβ1 treatment. Notably, knockdown of PFKFB3 substantially suppressed this induction (Figure 9A). Lactate, a downstream end product regulated by PFKFB3 in glycolysis and reported to activate renal fibroblasts [14], exhibited clear accumulation in response to TGFβ1, which was suppressed upon *PFKFB3* knockdown (Figure 9B). To verify the roles of lactate and PFKFB3 in fibroblast activation, we examined their effects on the expression of ACTA2, fibronectin, and PCNA. The induction of ACTA2, fibronectin, and PCNA in NRK-49F cells by TGFβ1 was significantly suppressed by *PFKFB3* knockdown. However, the suppression was reversed by lactate supplementation (Figure 9C,D).

## 4. Discussion

In this study, our main focus was to investigate how PFKFB3 mediates glycolysis to promote the activation of renal fibroblasts into myofibroblasts during fibrosis. Our results clearly demonstrate that glycolysis is enhanced in myofibroblasts, and this enhancement is mediated by the activation of the TGF-β signaling pathway, followed by the upregulation of PFKFB3. Consequently, the increased production of lactate, a product of glycolysis, directly contributes to fibroblast activation and proliferation, as evidenced by the increased expression of ACTA2, extracellular matrix components, and PCNA. The activation of fibroblasts may also have implications for renal tubular degeneration, suggesting a potential interplay between fibroblast activation and renal tubular damage in the context of fibrosis.

First, we investigated the metabolic changes occurring during fibroblast activation in renal fibrosis. Through the reanalysis of single-cell sequencing data from in vivo models of UUO and kidney ischemia/reperfusion, we observed the induction of gene expression related to the glycolysis pathway in myofibroblasts (Figure 2). Additionally, metabolomics testing revealed a significant accumulation of glycolytic metabolites, including lactate (Figure 9A,B). These findings collectively indicate a substantial induction of glycolysis during fibroblast activation in renal fibrosis. The induction of glycolysis in renal fibrosis has previously been observed in vitro using cultured renal fibroblasts treated with TGFβ1 [14], which aligns with our current findings. However, due to limitations in the number of fibroblast cells present in the kidney and the metabolic switch experienced by renal tubular cells from oxidative phosphorylation to glycolysis during renal fibrosis [12], it has been challenging to determine whether glycolysis is also enhanced in vivo during fibroblast activation. Our analysis of single-cell sequencing data represents the first evidence supporting enhanced glycolysis in myofibroblasts in an in vivo setting, highlighting the significance of this metabolic switch in the development of renal fibrosis.

In addition, we confirmed the novel role of PFKFB3 in renal fibrosis through the regulation of glycolysis. Inhibition of PFKFB3 in vitro resulted in the suppression of fibroblast activation and proliferation (Figure 8 to Figure 9), while myofibroblast-specific knockout of *Pfkfb3* significantly inhibited renal fibrosis induced by UUO and kidney ischemia/reperfusion in vivo (Figure 3, Figure 4, Figure 5 and Figure 6). Although we detected significant PFKFB3 upregulation after injury in the whole kidney, including the renal tubules (Figure 1), the results from the myofibroblast-specific *Pfkfb3*-knockout mouse model verified the pivotal role of PFKFB3 in fibroblast activation. However, the function of PFKFB3 in renal tubules remains unclear, which we aim to explore in our future study. Nevertheless, it is important to note that PFKFB3 is a dual-function protein, acting both as an enzyme to promote glycolysis and as a nuclear protein involved in cell cycle regulation [34]. In our previous study, we reported that PFKFB3 activation in renal tubules can enhance cisplatin nephrotoxicity through its interaction with CDK4, leading to the phosphorylation of Rb [23]. However, in this study, we clearly demonstrated that PFKFB3 is induced in the cytosol of renal fibroblasts following TGFβ1 treatment (Figure 7E), along with the induction of glycolysis (Figure 9A,B). Moreover, the restoration of the downstream glycolysis product lactate was able to reverse the effects of *PFKFB3* knockdown (Figure 9D). Therefore, in the context of renal fibrosis, PFKFB3 primarily regulates fibroblast activation through the promotion of glycolysis. This discovery of the new role of PFKFB3 also provides a potential therapeutic target for the treatment of renal fibrosis.

In renal fibroblast activation, multiple signaling pathways have been reported to play a role, including the TGFβ pathway, the HIF-1 pathway, the WNT/β-catenin pathway, the sonic Hedgehog pathway, and Notch pathways [33,35,36,37,38,39,40]. In our study, we observed a significant induction of PFKFB3 in the fibrotic kidney (Figure 1) or cultured renal fibroblasts with TGFβ1 stimulation (Figure 7, Figure 8 and Figure 9). However, we acknowledge that the regulation of PFKFB3 and glycolysis may also involve other signaling pathways. For instance, HIF-1 has been shown to regulate PFKFB3 expression [41]. Although HIF-1 activation in renal epithelial cells has been implicated in renal fibrosis [39,40] and epithelial PFKFB3 upregulation has been reported during renal fibrosis [12], it remains unclear whether there is fibroblast-specific HIF-1 activation that regulates PFKFB3. The WNT/β-catenin pathway has been associated with metabolic reprogramming in hepatic cancer cells [42], and Wnt3a has been shown to upregulate PFKFB3 during osteoblast differentiation in a β-catenin-independent manner [43]. Similarly, the sonic Hedgehog pathway can activate PFKFB3 in breast cancer cells. However, it is currently unknown whether these pathways are responsible for PFKFB3 upregulation in the kidney. Additionally, it has been reported that PFKFB3 activation may interact with or activate Notch signaling or counteract its function [19,44], although the regulatory mechanism between Notch and PFKFB3 expression remains unclear.

In the Pfkfb3^f/f^/Postn^MCM^ mouse model, the knockout of *Pfkfb3* in myofibroblasts was driven by the promoter of periostin, a protein exclusively expressed by myofibroblasts [6], and induced by tamoxifen treatment. However, due to the limited number of myofibroblasts in the kidney, especially in the early stages of fibrosis, it was challenging to precisely determine the starting time for *Pfkfb3* knockout. To ensure effective *Pfkfb3* knockout, we administered tamoxifen throughout the entire experimental procedure. Because tamoxifen has been reported to be anti-fibrotic in the kidney [45,46], all mice received the same dosage of tamoxifen to ensure consistent and comparable responses. While it has been reported that early fibroblast activation after injury can be protective and promote renal tubular repair [47], excessive fibrosis in the later stage can lead to scarring. Despite this, we observed significant suppression of renal fibrosis by *Pfkfb3* knockout (Figure 3, Figure 4, Figure 5 and Figure 6), indicating that the later stage of scarring was effectively suppressed.

Indeed, the suppressed renal fibrosis observed in the *Pfkfb3*^f/f^/*Postn*^MCM^ kidneys also contributed to better renal tubular degeneration and repair. Notably, kidney atrophy was significantly attenuated in these mice (Figure 3C,D and Figure 5C,D), indicating that *Pfkfb3* knockout has a protective effect on the kidney size. Histological examination further revealed better preservation of renal tubular structures (Figure 4A and Figure 6A), suggesting that the absence of *Pfkfb3* in myofibroblasts leads to less renal tubular damage. Additionally, the reduced Vimentin induction observed could be attributed to either a decrease in myofibroblasts or less renal tubular degeneration. These results collectively indicate that myofibroblast-specific *Pfkfb3* knockout leads to less renal stiffness and, importantly, has a protective feedback effect on renal tubules.

## 5. Conclusions

In conclusion, our study provides compelling evidence that PFKFB3 plays a crucial role in the development of renal fibrosis. Following UUO or ischemia/reperfusion injury, PFKFB3 is induced, leading to an enhancement of glycolysis in the kidney. The subsequent accumulation of lactate promotes fibroblast activation and contributes to excessive renal fibrosis. *Pfifb3* knockout in fibroblasts leads to a significant reduction in the fibrotic response, scarring, and maladaptive tubular repair. These findings highlight the importance of PFKFB3 and glycolysis in renal fibrosis and suggest that targeting PFKFB3 may represent a potential therapeutic approach to mitigate fibrosis and promote better renal outcomes.

## Figures and Tables

**Figure 1 cells-12-02081-f001:**
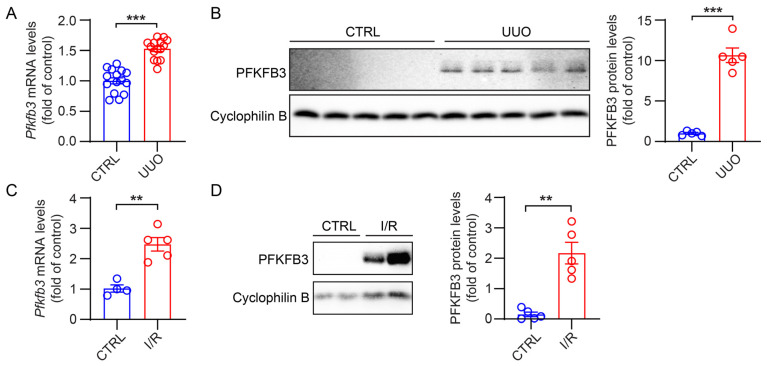
The increased PFKFB3 expression in fibrotic kidneys. (**A**) qRT-PCR analysis of the mRNA expression of *Pfkfb3* in the unilateral ureteral obstruction (UUO) and control kidneys (n = 15). (**B**) Representative Western blots and their quantification showing PFKFB3 protein levels in the UUO and control kidneys (n = 5). (**C**) qRT-PCR analysis of the mRNA expression of *Pfkfb3* in the ischemia/reperfusion (I/R) and control kidneys (n = 4 for CTRL, n = 5 for I/R). (**D**) Representative Western blots and their quantification showing PFKFB3 protein levels in the I/R and control kidneys (n = 5). Data are means ± SEM. ns, no significance; ** *p* < 0.01; *** *p* < 0.001 for indicated comparisons. Statistical significance was determined using Student’s *t*-test.

**Figure 2 cells-12-02081-f002:**
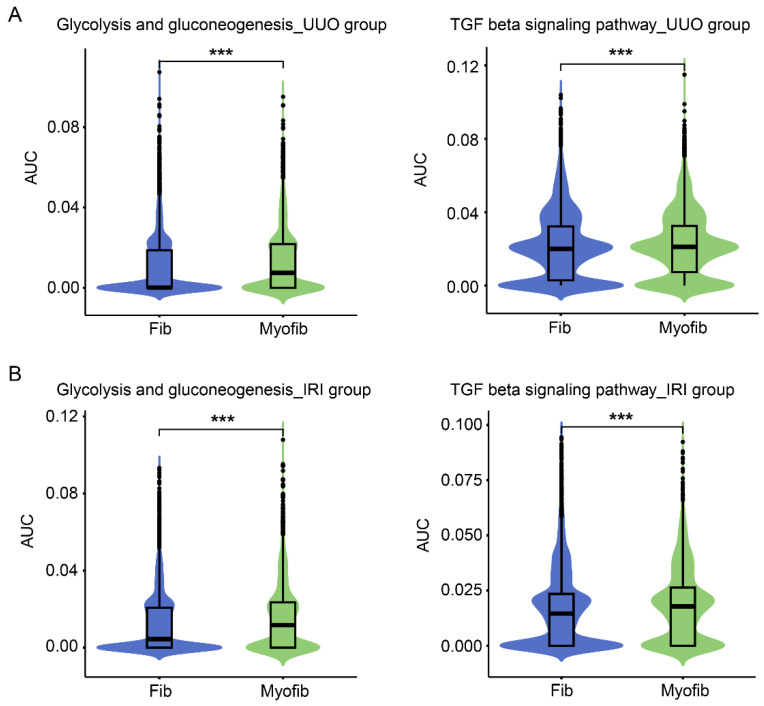
scRNAseq reveals upregulated glycolysis in myofibroblast cells compared to fibroblast cells in the UUO and IRI kidneys. (**A**) Violin plots of glycolysis and gluconeogenesis pathways and TGFβ signaling in fibroblast and myofibroblast clusters in the kidney following the UUO model. (**B**) Violin plots of glycolysis and gluconeogenesis pathways and TGFβ signaling in fibroblast and myofibroblast clusters in the kidney following the IRI model. Data are means ± SEM. *** *p* < 0.001 for indicated comparisons. Statistical significance was determined using Student’s *t*-test.

**Figure 3 cells-12-02081-f003:**
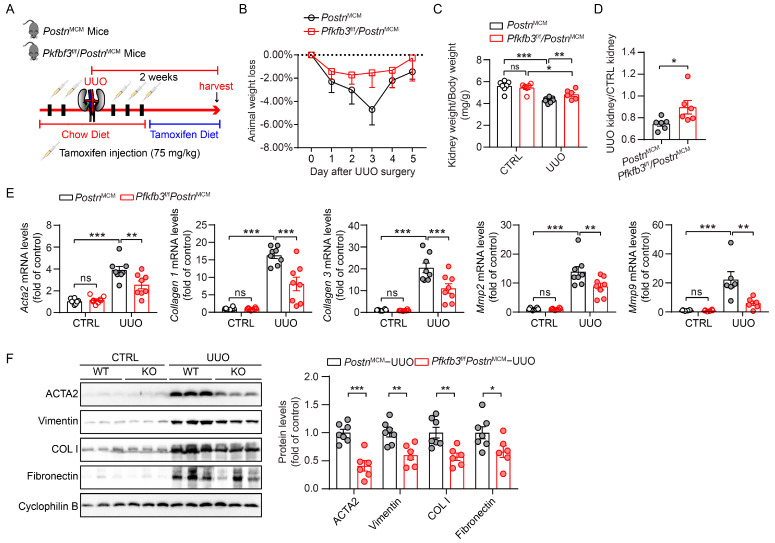
Myofibroblast *Pfkfb3* deficiency suppresses renal fibrosis in the UUO mouse model. (**A**) Schematic illustration of UUO in myofibroblast *Pfkfb3*-deficient mice. (**B**) The daily body weight loss percentage for *Pfkfb3*^f/f^*/Postn*^MCM^ and *Postn*^MCM^ mice subjected to UUO surgery for 5 days. (**C**) The ratio of kidney weight to body weight for *Pfkfb3*^f/f^*/Postn*^MCM^ and *Postn*^MCM^ mice subjected to UUO surgery for 14 days (n = 7 *Postn*^MCM^ mice, n = 6 *Pfkfb3*^f/f^*/Postn*^MCM^ mice). (**D**) The ratio of fibrotic kidney weight to control kidney weight for *Pfkfb3*^f/f^*/Postn*^MCM^ and *Postn*^MCM^ mice subjected to UUO surgery for 14 days (n = 6 mice per group). (**E**) qRT-PCR analysis of the mRNA expression of *Acta2*, *Col1*, *Col3*, *Mmp2*, and *Mmp9* in kidneys collected from *Pfkfb3*^f/f^*/Postn*^MCM^ and *Postn*^MCM^ mice at day 14 post-UUO surgery (n = 8 mice/group). (**F**) Representative Western blots and their quantification showing ACTA2, Vimentin, COL I, and FN protein expression in kidneys collected from *Pfkfb3*^f/f^*/Postn*^MCM^ and *Postn*^MCM^ mice at day 14 post-UUO surgery (n = 7 *Postn*^MCM^ mice, n = 6 *Pfkfb3*^f/f^*/Postn*^MCM^ mice). Data are means ± SEM. ns, no significance; * *p* < 0.05; ** *p* < 0.01; *** *p* < 0.001 for indicated comparisons. Statistical significance was determined using one-way ANOVA followed by the Bonferroni test (**C**,**E**) or using Student’s *t*-test (**D**,**F**).

**Figure 4 cells-12-02081-f004:**
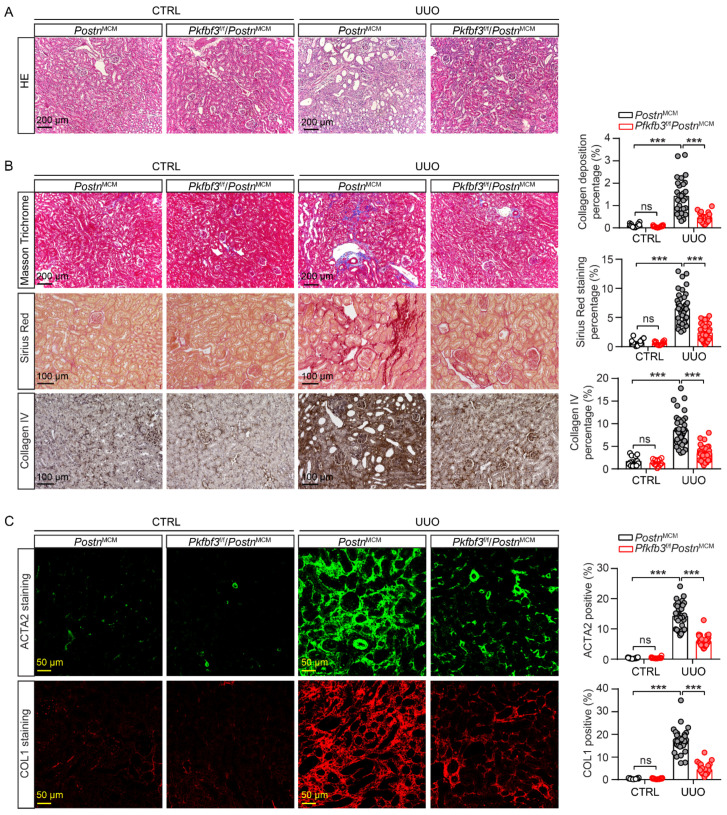
Myofibroblast *Pfkfb3* deficiency suppresses renal fibrosis in the UUO mouse model. *Pfkfb3*^f/f^*/Postn*^MCM^ and *Postn*^MCM^ mice were subjected to UUO for 14 days, and the kidneys were collected and fixed for paraffin-embedded sections. (**A**) Representative image of hematoxylin and eosin (H&E) staining (n = 9 mice/group). (**B**) Representative image and quantification data of Masson’s trichrome staining, Sirius Red staining, and COL IV staining. (**C**) Representative image and quantification data of ACTA2 and COL1 immunostaining. Scale bar = 200 μm, 100 μm, or 50 μm (n = 3 mice for the control group, n = 9 mice for the UUO group, 4 areas/section were quantified). Data are means ± SEM. ns, no significance; *** *p* < 0.001 for indicated comparisons. Statistical significance was determined using one-way ANOVA followed by the Bonferroni test.

**Figure 5 cells-12-02081-f005:**
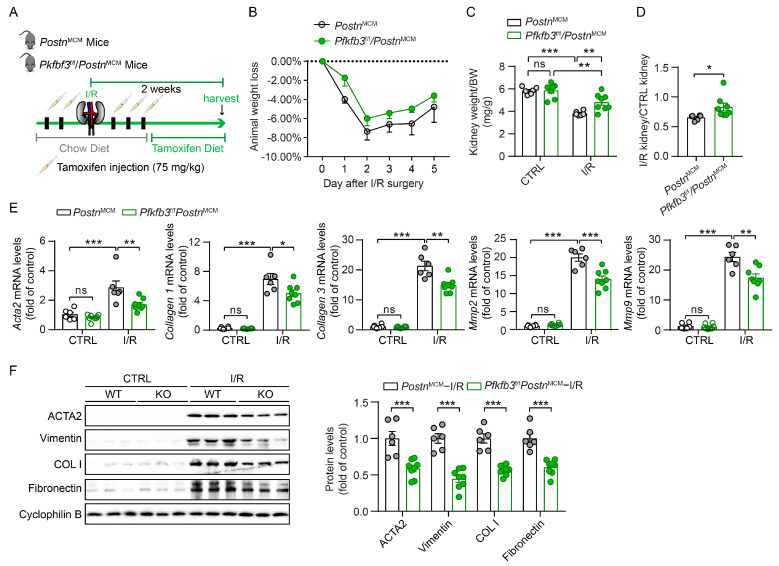
Myofibroblast *Pfkfb3* deficiency suppresses renal fibrosis in the ischemia/reperfusion mouse model. (**A**) Schematic illustration of I/R in myofibroblast *Pfkfb3*-deficient mice. (**B**) The daily body weight loss percentage for *Pfkfb3*^f/f^*/Postn*^MCM^ and *Postn*^MCM^ mice subjected to I/R surgery for 5 days. (**C**) The ratio of kidney weight to body weight for *Pfkfb3*^f/f^*/Postn*^MCM^ and *Postn*^MCM^ mice subjected to I/R surgery for 14 days. (**D**) The ratio of fibrotic kidney weight to control kidney weight for *Pfkfb3*^f/f^*/Postn*^MCM^ and *Postn*^MCM^ mice subjected to I/R surgery for 14 days. (**E**) qRT-PCR analysis of the mRNA expression of *Acta2*, *Col1*, *Col3*, *Mmp2*, and *Mmp9* in kidneys collected from *Pfkfb3*^f/f^*/Postn*^MCM^ and *Postn*^MCM^ mice at day 14 post-I/R surgery. (**F**) Representative Western blots and their quantification showing ACTA2, Vimentin, COL I, and FN protein expression in kidneys collected from *Pfkfb3*^f/f^*/Postn*^MCM^ and *Postn*^MCM^ mice at day 14 post-I/R surgery (n = 6 *Postn*^MCM^ mice, n = 8 *Pfkfb3*^f/f^*/Postn*^MCM^ mice). Data are means ± SEM. ns, no significance; * *p* < 0.05; ** *p* < 0.01; *** *p* < 0.001 for indicated comparisons. Statistical significance was determined using one-way ANOVA followed by the Bonferroni test (**C**,**E**) or using Student’s *t*-test (**D**,**F**).

**Figure 6 cells-12-02081-f006:**
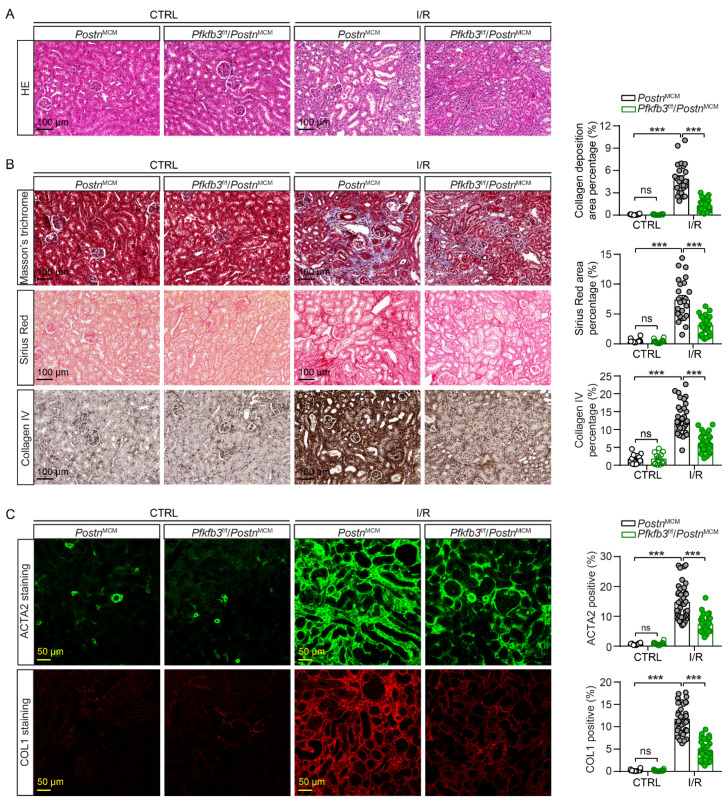
Myofibroblast *Pfkfb3* deficiency suppresses renal fibrosis in the ischemia/reperfusion mouse model. *Pfkfb3*^f/f^*/Postn*^MCM^ and *Postn*^MCM^ mice were subjected to I/R for 14 days, and the kidneys were collected and fixed for paraffin-embedded sections. (**A**) Representative image of hematoxylin and eosin (H&E) staining. (**B**) Representative image and quantification data of Masson’s trichrome staining, Sirius Red staining, and COL IV staining. (**C**) Representative image and quantification data of ACTA2 and COL1 immunostaining. Scale bar = 100 μm or 50 μm (n = 3 mice for the control groups, n = 6 *Postn*^MCM^ I/R mice, n = 8 *Pfkfb3*^f/f^*/Postn*^MCM^ I/R mice, 4 areas/section were quantified). Data are means ± SEM. ns, no significance; *** *p* < 0.001 for indicated comparisons. Statistical significance was determined using one-way ANOVA followed by the Bonferroni test.

**Figure 7 cells-12-02081-f007:**
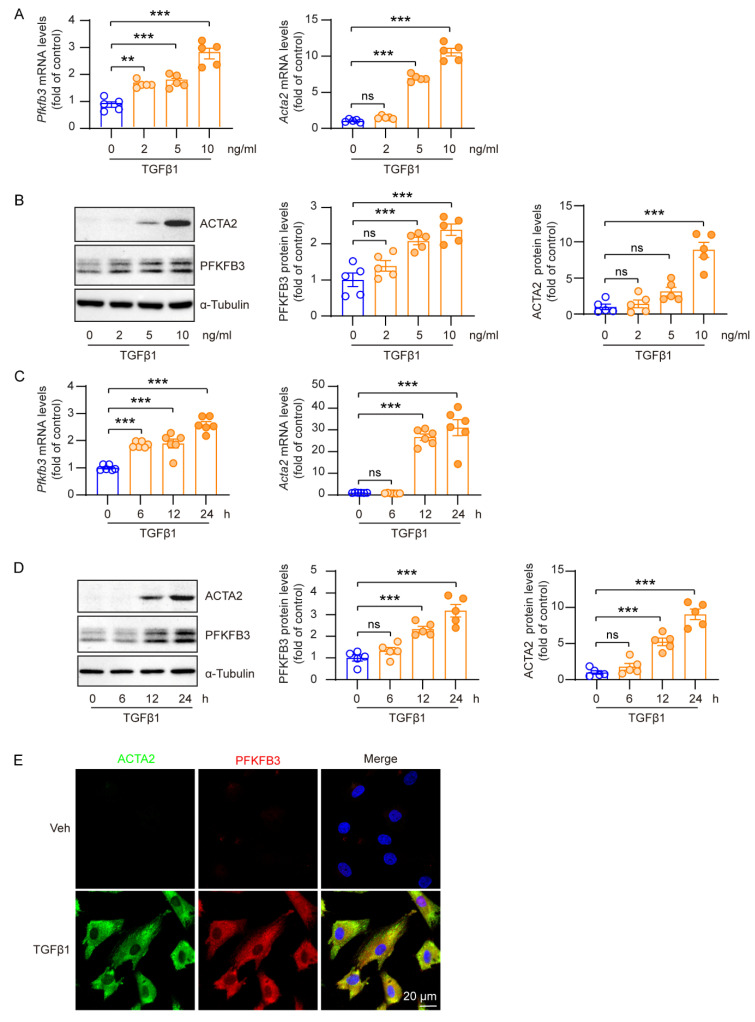
Increased PFKFB3 expression in TGFβ1-activated renal fibroblasts. (**A**) qRT-PCR analysis of the mRNA expression of *Pfkfb3* and *Acta2* in NRK-49F cells treated with TGFβ1 or a vehicle for 24 h at different dosages (n = 5). (**B**) Representative Western blots and their quantification showing PFKFB3 and ACTA2 protein levels in NRK-49F cells treated with TGFβ1 or a vehicle for 24 h at different dosages (n = 5). (**C**) qRT-PCR analysis of the mRNA expression of *Pfkfb3* and *Acta2* in NRK-49F cells treated with TGFβ1 (10 ng/mL) or a vehicle for different times (n = 6). (**D**) Representative Western blots and their quantification showing PFKFB3 and ACTA2 protein levels in NRK-49F cells treated with TGFβ1 (10 ng/mL) or a vehicle for different times (n = 5). (**E**) Representative image of ACTA2 and PFKFB3 staining of NRK-49F cells treated with TGFβ1 (10 ng/mL) for 24 h. Scale bar = 20 μm. Data are means ± SEM. ns, no significance; ** *p* < 0.01; *** *p* < 0.001 for indicated comparisons. Statistical significance was determined using one-way ANOVA followed by the Bonferroni test (**A**–**D**).

**Figure 8 cells-12-02081-f008:**
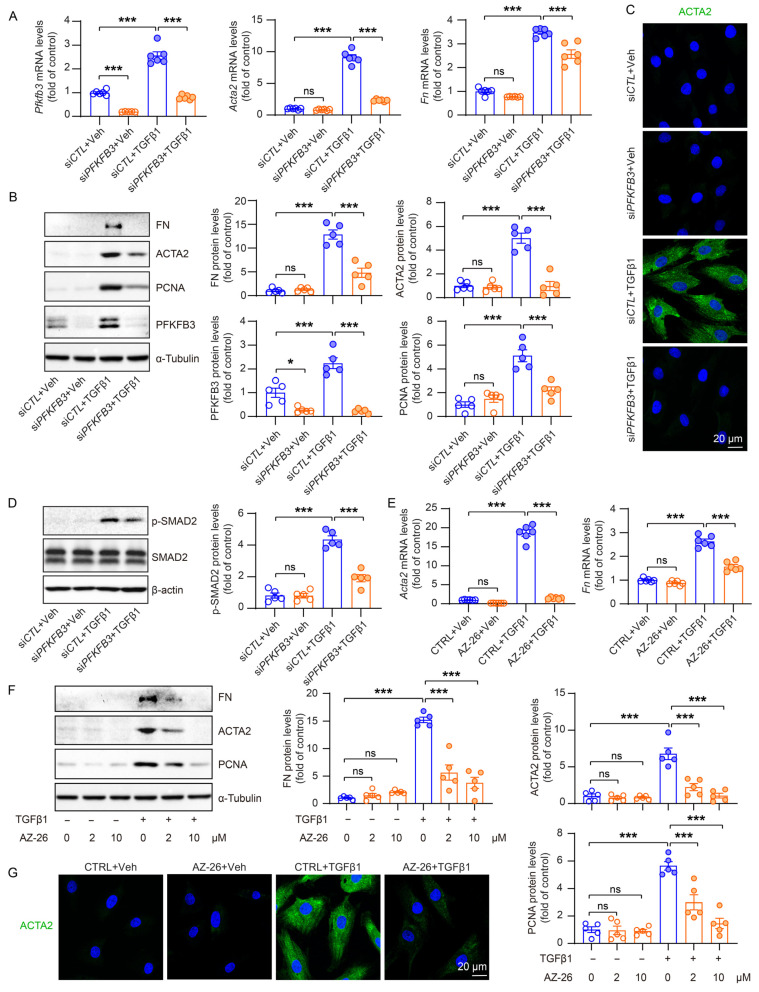
The reduced fibrotic activity of Pfkfb3-knockdown renal myofibroblasts. (**A**) qRT-PCR analysis of the mRNA expression of *Pfkfb3*, *Acta2*, and *Fn* in NRK-49F cells transfected with si*CTL* or si*PFKFB3* for 48 h and treated with a vehicle or TGFβ1 for 24 h; n = 6. (**B**) Representative Western blots and their quantification showing FN, ACTA2, PCNA, and PFKFB3 protein levels in NRK-49F cells transfected with si*CTL* or si*PFKFB3* for 48 h and treated with a vehicle or TGFβ1 for 24 h; n = 5. (**C**) Representative image of ACTA2 staining of NRK-49F cells transfected with si*CTL* or si*PFKFB3* for 48 h and treated with a vehicle or TGFβ1 for 24 h. Scale bar = 20 μm. (**D**) Representative Western blots and quantification of *p*-smad2, smad2/3, and ACTA2 in NRK-48F cells transfected with si*CTL* or si*PFKFB3* for 48 h and treated with a vehicle or TGFβ1 for 24 h. (**E**) qRT-PCR analysis of the mRNA expression of *Acta2* and *Fn* in NRK-49F cells treated with a vehicle or TGFβ1 for 24 h in the absence or presence of AZ-26 (n = 6). (**F**) Representative Western blots and their quantification showing FN, ACTA2, and PCNA protein levels in NRK-49F cells treated with a vehicle or TGFβ1 for 24 h in the absence or presence of AZ-26 (n = 5). (**G**) Representative image of ACTA2 staining of NRK-49F cells treated with a vehicle or TGFβ1 for 24 h in the absence or presence of AZ-26. Scale bar = 20 μm. Data are means ± SEM. ns, no significance; * *p* < 0.05; *** *p* < 0.001 for indicated comparisons. Statistical significance was determined using one-way ANOVA followed by the Bonferroni test (**A**,**B**,**D**–**F**).

**Figure 9 cells-12-02081-f009:**
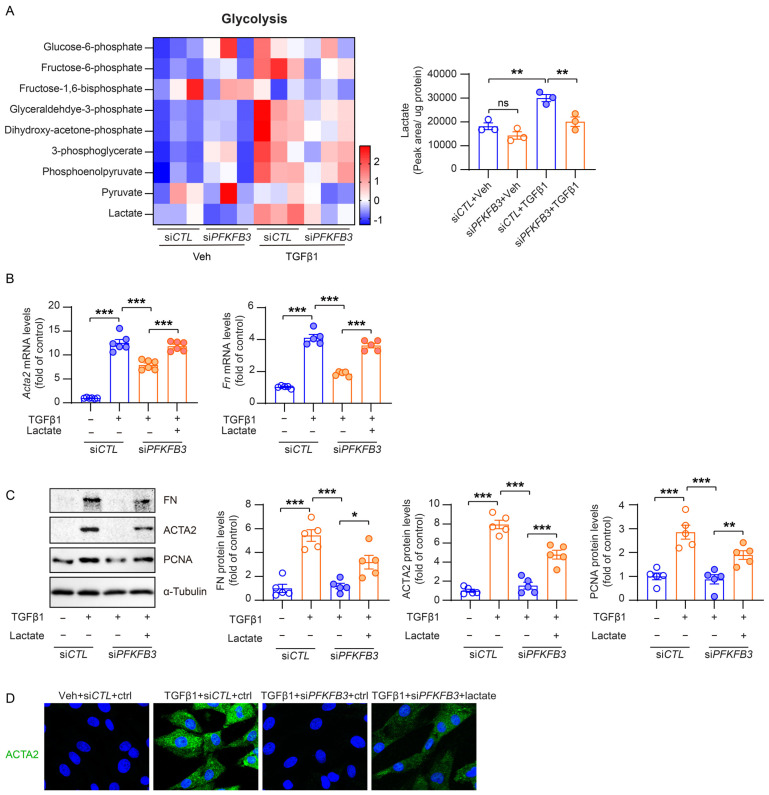
Decreased glucose metabolism in *Pfkfb3*-knockdown renal myofibroblasts. (**A**) Heat map showing the metabolites in the glycolysis pathway and quantification data of lactate levels in NRK-49F cells treated with TGFβ1 (10 ng/mL) for 24 h (n = 3). (**B**) qRT-PCR analysis of the mRNA expression of *Acta2* and *Fn* in NRK-49F cells transfected with si*CTL* or si*PFKFB3* for 48 h and treated with a vehicle or TGFβ1 for 24 h in the absence or presence of lactate (5 mM) (n = 5). (**C**) Representative Western blots and their quantification showing FN, ACTA2, PCNA, and PFKFB3 protein levels in NRK-49F cells transfected with si*CTL* or si*PFKFB3* for 48 h and treated with a vehicle or TGFβ1 for 24 h in the absence or presence of lactate (5 mM) (n = 5). (**D**) Representative image of ACTA2 staining of NRK-49F cells transfected with si*CTL* or si*PFKFB3* for 48 h and treated with a vehicle or TGFβ1 for 24 h in the absence or presence of lactate (5 mM). Scale bar = 20 μm. Data are means ± SEM. ns, no significance; * *p* < 0.05; ** *p* < 0.01; *** *p* < 0.001 for indicated comparisons. Statistical significance was determined using one-way ANOVA followed by the Bonferroni test (**A**–**C**).

## Data Availability

The metabolomics data have been uploaded to MetaboLights (MTBLS8281).

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
