# Peer review of "PFKFB3-Mediated Glycolysis Boosts Fibroblast Activation and Subsequent Kidney Fibrosis"

_cells, 2023, doi:10.3390/cells12162081_

Round 1
Reviewer 1 Report
This interesting manuscript performed by Dr. Yang and co-authors systemically investigated the role of myofibroblast-derived Pfkfb3 in determining the extent of kidney fibrosis through impact glycolysis. The overall experimental design was beautiful. The authors employed public data mining, bioinformatics, conditional knockout mice, and multiple CKD in vivo and in vitro models for the current study. The presented data is sufficient and graceful, and I believe it perfectly fits the Cells journal. Currently, energy metabolism is actually a hot point in the field of kidney diseases and elucidating kidney fibrosis from the perspective of myofibroblast metabolism is very novel. Therefore, it is the right time to publish this manuscript. Just one minor comment:
1. Figure 1, panel C: Pfkfb3 mRNA data in the IRI model was missed. The representative western blot bands and quantitative data should be panel D based on the figure legend.
Author Response
Response to Review 1:
Reviewer's concern: This interesting manuscript performed by Dr. Yang and co-authors systemically investigated the role of myofibroblast-derived Pfkfb3 in determining the extent of kidney fibrosis through impact glycolysis. The overall experimental design was beautiful. The authors employed public data mining, bioinformatics, conditional knockout mice, and multiple CKD in vivo and in vitro models for the current study. The presented data is sufficient and graceful, and I believe it perfectly fits the Cells journal. Currently, energy metabolism is actually a hot point in the field of kidney diseases and elucidating kidney fibrosis from the perspective of myofibroblast metabolism is very novel. Therefore, it is the right time to publish this manuscript. Just one minor comment:
Figure 1, panel C: Pfkfb3 mRNA data in the IRI model was missed. The representative western blot bands and quantitative data should be panel D based on the figure legend.
Response: We are grateful for the reviewer’s comments. We have included the Pfkfb3 qPCR data in Figure 1 and revised the description in Results and Figure Legend accordingly (Page 6).
Reviewer 2 Report
1. What is the main question addressed by the research?
The critical role of PFKFB3 (an activator of glycolysis) in driving fibroblast activation and subsequent renal fibrosis
2. Do you consider the topic original or relevant in the field? Does it
address a specific gap in the field?
Yes
3. What does it add to the subject area compared with other published
material?
The study provides compelling evidence that PFKFB3 plays a crucial role in the development of renal fibrosis
4. What specific improvements should the authors consider regarding the
methodology? What further controls should be considered?
The experiments were well designed and done. I have a minor suggestion to authors. While the authors convincingly demonstrated that PFKFB3 is the critical downstream target of TGFbeta1 for its induction of renal fibroblast activation, I am wondering whether PFKFB3 activation will also enhance TGFbeta1 signaling.
5. Are the conclusions consistent with the evidence and arguments presented
and do they address the main question posed?
Yes
6. Are the references appropriate?
yes
7. Please include any additional comments on the tables and figures.
N/A
Author Response
Response to Reviewer 2:
Reviewer's concerns:
What specific improvements should the authors consider regarding the
methodology? What further controls should be considered?
The experiments were well designed and done. I have a minor suggestion to authors. While the authors convincingly demonstrated that PFKFB3 is the critical downstream target of TGFbeta1 for its induction of renal fibroblast activation, I am wondering whether PFKFB3 activation will also enhance TGFbeta1 signaling.
Response:
Thank you for your insightful question. To assess TGFb1 signaling activation, we conducted immunoblots to analyze the phosphorylation status of Smad2. We observed a decrease in Smad2 phosphorylation following PFKFB3 knockdown in NRK-48F cells. These findings are presented in Figure 8D (Page 16), and we have adjusted the corresponding text description accordingly (Page 15).
Reviewer 3 Report
Yang et al. report on the role of PFKFB3 in renal fibrosis. Their mechanisms are related to the glycolysis in the renal fibroblast by regulation of PFKFB3. The authors use two different fibrosis models, UUO and Unilateral IR injury model, clearly showing the results.
I have minor comments on this work.
1. Authors focus on the role of PFKFB3 in fibroblast and myofibroblast during renal fibrosis. However, most of the cell populations of the kidney are proximal tubules. Therefore, the authors need additional comments on the role of PFKFB3 in tubule cells.
2. Authors use Pfkfb3f/f/PostnMCM mouse model for myofibroblast-specific Pkfkb2 knockout, which was induced by tamoxifen treatment. Is there any possibility for tamoxifen treatment to influence fibroblast activation? Several reports have published the protective effects of tamoxifen on UUO-induced renal fibroblast activation and fibrosis.
Author Response
Response to Reviewer 3:
Reviewer's concern:
Yang et al. report on the role of PFKFB3 in renal fibrosis. Their mechanisms are related to the glycolysis in the renal fibroblast by regulation of PFKFB3. The authors use two different fibrosis models, UUO and Unilateral IR injury model, clearly showing the results.
I have minor comments on this work.
- Authors focus on the role of PFKFB3 in fibroblast and myofibroblast during renal fibrosis. However, most of the cell populations of the kidney are proximal tubules. Therefore, the authors need additional comments on the role of PFKFB3 in tubule cells.
Response:
Thank you for highlighting the role of PFKFB3 in renal tubules. In our study, we identified a significant induction of PFKFB3 in the whole kidney after injury, including the renal tubules (Fig. 1 and data not shown). Currently, the role of PFKFB3 in renal tubules is unclear which we aim to explore in our future study. In the revised manuscript, we have included a statement about the role of renal tubule PFKFB3 in discussion (Page 20).
Reviewer's concern:
- Authors use Pfkfb3f/f/PostnMCMmouse model for myofibroblast-specific Pkfkb2knockout, which was induced by tamoxifen treatment. Is there any possibility for tamoxifen treatment to influence fibroblast activation? Several reports have published the protective effects of tamoxifen on UUO-induced renal fibroblast activation and fibrosis.
Response:
Thank you for drawing attention to the effect of tamoxifen. Indeed, tamoxifen has been reported to protect the kidneys from fibrosis in UUO and other chronic injury conditions. To ensure uniformity in the effect and response to tamoxifen across our study, we administered the same dosage to all the mice involved in the experiments, as detailed in the Methods section. We observed no significant weight loss differences among the mice after they recovered from surgery (Fig. 3B, Fig. 5B). Thus, we assume that they consumed similar amount of tamoxifen-containing mouse chow thereafter. In the revised manuscript, we have added a sentence in the discussion to emphasize the role and effect of tamoxifen in our study (Page 21).